# Neural Heterogeneous Scheduler

**Tegg Taekyong Sung** [1] [*]   **Valliappa Chockalingam** [1] [2] [*]   **Alex Yahja** [1]   **Bo Ryu** [1]

## Abstract

Access to massive computation allows researchers and developers to succeeded in using technology to enhance processes in many applications. However there have been claims as to the tapering of the exponential decrease in the cost of hardware (following Moore's law) due to physical hardware limitations. Next generation special purpose systems making using of multiple kinds of co-processors, known as heterogeneous system-on-chips, have been in active research recently. In this paper, we introduce a method to intelligently schedule a stream of tasks to available processing elements in such a system. We use deep reinforcement learning which allows for complex decision making and demonstrate that machine learning can be used for scheduling decisions and provides for a viable, likely better alternative to reducing execution time, given a set of tasks.

## 1. Introduction

Numerous breakthroughs in deep learning would not be possible without the help of hardware capable of massive amounts of computation. Especially in machine learning, neural networks have taken advantage of parallelization through GPUs and, more recently, special purpose hardware like TPUs have been used to perform millions of computations in a quicker manner. However, when we consider recent real-world applications built with embedded hardware, different considerations have to be made. For instance, custom ASICs and FPGAs have different performance capabilities and power and energy consumption. So, essential care must be taken in choosing which applications to develop on which types of hardware. Concurrently, hardware performance has been becoming cheaper, following Moore's

law (Schaller, 1997), but transistor density in microprocessors has also recently reached the maximum projection based on physical limitations.

The System-on-Chip (SoC) architecture has been revealed as a novel approach to chip design which merges different levels of computational cores (Borel, 1997). In addition, domain-specific heterogeneous SoCs enable different functionalities necessary in certain domains and provide for easy software implementations. Heterogeneous chips have the strength that they can attain the best performance for certain applications if the cores in the chip are systematically scheduled once the task becomes available. Combinations of operations are optimally scheduled to process jobs with different requirements. Heterogeneous processors are expected to break traditional trade-offs such as that between power and performance. Thus, how to optimally schedule operations becomes a main research topic which is generally known as an NP-hard problem. While there are many heuristic or approximate algorithms that can make this problem more tractable, we take the view that optimally distributing ready-be-assigned tasks into available processing elements in the heterogeneous SoC can be formalized as a sequential decision making problem.

In this paper, we use deep reinforcement learning (DRL) which provides for a powerful and flexible way to solve complex sequential decision making problems. Here, we especially consider tasks that have dependencies. Thus, the scheduling agent must learn how to schedule the tasks given that some tasks may require other tasks to have already run. This makes the problem difficult due to long-term dependencies and partial observability. Moreover, without pre-emptions, the former entails that the agent cannot choose scheduling actions at every time step but only when assigned tasks are completed and new tasks ready to be scheduled appear. The following sections describe the details of the simulation environment which takes a job consisting of a set of tasks and a resource matrix specifying the performance and communication specifications of the processing elements. Next, we formalize the reinforcement learning (RL) setting and describe the policy-based algorithm we use to tackle the sequential decision-making problem. In the experiments, we compare our model, which we interchangeably refer to as Deep Resource Manager (DRM) or Neural Heterogeneous Scheduler, with baselines to verify that the

---

[*]Equal contribution  [1]EpiSys Science, San Diego, USA [2]Department of Computer Science, University of Alberta, Edmonton, Canada.  Correspondence to: Tegg Taekyong Sung <tegg@episyscience.com>, Valliappa Chockalingam <valli@episyscience.com>.

*Proceedings of the 36th International Conference on Machine Learning*, Long Beach, California, PMLR 97, 2019. Copyright 2019 by the author(s).

performance of the Deep RL agent we introduce is better than different heuristic scheduling algorithms. We also provide saliency map and GANTT chart visualizations during the learning process to reason about the agent's decisions.

## 2. Background

Deep reinforcement learning (Deep RL) has been successfully applied to several domains such as robotics (Levine et al., 2016) and games (Silver et al., 2017; Vinyals et al., 2019). Most successful RL applications stand in the usual RL framework of Markov decision processes. However, in our case, actions can take various amounts of time to complete. Scheduling chip processors in real-world applications involves a continually filling stream of tasks where many activities progress simultaneously. Action decisions are only performed when tasks are ready to be scheduled. Given these properties and limitations of the environment, this process can be defined as semi-Markov decision process (SMDP) with temporally-extended actions or options. When an assigned action is not completed, then the agent essentially takes the 'no-operation' action.

Mathematically, the MDP setting can be formalized as a 5-tuple $\langle \mathcal{S}, \mathcal{A}, R, P, \gamma \rangle$ (Sutton & Barto, 2018; Puterman, 2014). Here, $S$ denotes the state space, $A$, the action space, $R$, the reward signal which is generally defined over states or state-action pairs, $P$, a stochastic matrix specifying transition probabilities to next states given a state and action, and $\gamma \in [0, 1]$, the discount factor.

Normally, the SMDP framework would involve an option framework, augmenting the action space, but instead, we use simple options here that take no-op actions and hence leave the option framework with preemption of running tasks as future work.

In addition, the heterogeneous resource management environment is essentially partially observable, because the agent can only observe the tasks ready-to-be-assigned to a processing element. To address this, we augment the state with the other task lists as well (not just the ready list containing the ready-to-be-assigned tasks) and transition to fully-observable problem.

## 3. Proposed Approach

### 3.1. Environment Setting

The heterogeneous SoC chip we consider is to be used in various applications such as WiFi RX/TX or pulse doppler is simulated in a realistic discrete-event environment, DASH-Sim, which is described in 3.1.1. It is developed with the SimPy library (Lnsdorf & Scherfke, 2018) to implement running tasks in a continuous time frame setting. The specifications of the set of tasks and processing elements are

written within `job` and `resource_matrix` files that are described in 3.1.2.

### 3.1.1. SIMULATION

There are some manufactured embedded boards like Zynq A53, Odroid A7, or Odroid A15 with their heterogeneous chip sets, and their functionalities include encoder and decoder, scrambler and descrambler, interleaver and deinterleaver, QPSK modulation and demodulation, FFT and inverse-FFT. We however consider simulation as in many RL applications. The goal of an agent is to achieve a low time to completion, given task combinations.

In the recent past, RL algorithms have been commonly developed using the OpenAI Gym environment interface (Brockman et al., 2016) and many assume the Markov decision process framework, whereas we use SimPy environment, which simulates sequential discrete-events. Each event represents a task. Each task, upon execution, runs till completion and the scheduler can only choose to schedule tasks in the ready list. The simulator is visualized in Figure 1.

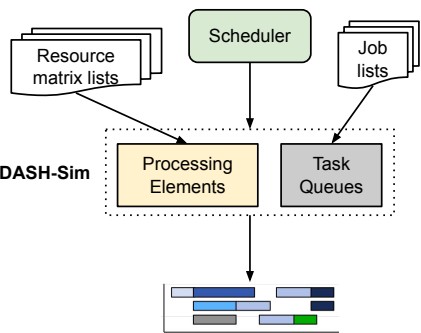

*Figure 1.* An overall diagram of the DASH-Sim environment which runs with the SimPy discrete-event library. The simulator constructs task queues and processing elements based on the descriptions in the `job` and `resource_matrix` file lists. The scheduler assigns tasks in the ready list to cores in SoC chip.

Prior to running a simulation, the information of processing elements (PEs) and tasks are parsed with `resource_matrix` and `job` text files described in 3.1.2. Each PE represents chipsets such as RAM, CPU, GPU, or memory accelerators in heterogeneous SoC and have different execution time, energy and power consumption. In this paper, we only consider the execution time for the performance.

At the start of a job, all the tasks are fed into the *outstanding list* and the ones that do not have task dependencies are pushed into the *ready list*. Then, the agent assigns a processing element, through an ID, `PE_ID`, to tasks that are in the *ready list* and these tasks then proceed to the *running*

*list* when the begin execution. During this time period, the agent chooses a 'no-operation' action for the running tasks. Once a task completes, it is moved to the *completed list*. If all the tasks are in the *completed list*, the scheduling episode is finished and the next `resource_matrix` and `job` files in the list are used for the next episode.

### 3.1.2. TASK AND RESOURCE MATRIX

Tasks are constantly generated and a scheduler distributes them to different chipsets in the SoC. We assume tasks have dependencies such as those shown in Figure 2. We describe the list of tasks in a `job` file and the associated processing elements in a `resource_matrix` file. Their structures are described in below.

**Job list**

- `job_name` <job name>

- `add_new_tasks` <number of tasks>

- <task name> <task ID> <task predecessors>

- <task name> <earliest start time> <deadline>

**Resource matrix list**

- `add_new_resource` <resource ID> <number of tasks>

- <task name> <performance>

In a job file, tasks have `HEAD` and `TAIL` flags that indicate the start and end. In this paper, we consider 10 tasks of one job with 3 processing elements. However, we add randomization to the resource matrix to train our agent be more robust. This results in differing performances. Performance here refers to the fact that the execution time taken to process a given task in a given processing element varies. The earliest start time, deadline, and performance are all given in units of milliseconds.

### 3.2. Algorithm

In this paper, we develop a new agent using deep reinforcement learning to allocate resources to a heterogeneous SoC, a long-term credit assignment task. As described in the Section 2, the environment in its most general form can be thought of as a partially observable SMDP.

Figure 3 shows the interaction between DASH-Sim and DRM scheduler. Task list transitions are controlled by DASH-Sim environment. The scheduler agent takes the tasks from the ready list as an input and assigns each task a `PE_ID`. Particularly, our DRM scheduler receives ready tasks but also generates state representations with all the

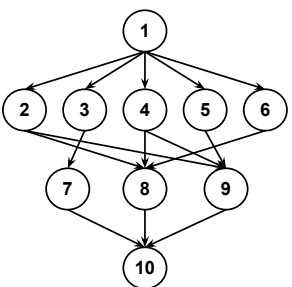

*Figure 2.* Task dependency visualization of the `job_Top` file. Each circle represents task numbers and arrows show task dependencies.

task lists. We convert the task lists and resource matrix to binary vector representations when representing integer values, multi-binary representations for state features that can take on multiple values and concatenate the represenation to form the final state representation vector. This information from all the state lists not only addresses partial observability in the DASH-Sim environment, but also gives additional information about the relations between tasks through the task list transitions.

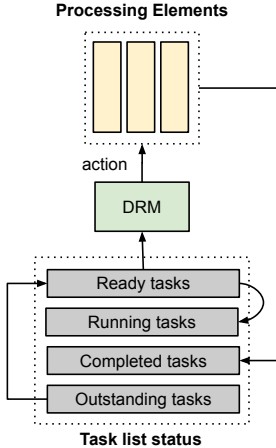

*Figure 3.* A diagram of the interaction of DRM architecture and DASH-Sim environment. Tasks in the lists are controlled by DASH-Sim environment and DRM assigns every `PE_ID` to ready tasks.

We use an actor-critic algorithm described in Equation 1. The agents action is taken for tasks in ready list. We use a simple reward of -1 per timestep to encourage the agent to complete tasks quickly. At the end of the episode, the agent is updated by looking over the past state representation when action choices were made and the resulting discounted reward for the scheduling decision made. These updates follow the traditional actor-critic losses and happen on-policy as the function approximator is not updated

during an episode. Additionally, we use a decaying temperature in the SoftMax action selection to gradually reduce exploration and move between SoftMax to argmax. This is similar to the addition of entropy and lets the agent avoid skewed action distribution early in learning and introduces more exploration at the beginning of the simulation while slowly relying on exploitation later in training.

$$\nabla_\theta J(\theta) = \mathbb{E}_{\pi_\theta}[\nabla_\theta \log \pi_\theta(a_t|s_t)A_t(s_t)], \qquad (1)$$

Above, the objective is to find $\theta$ that parameterizes the neural network to maximize $J$. $A_t$ is the advantage that subtracts the state-value of state $s_t$, $V(s_t)$, from $G_t$, the empirically observed discounted reward, the baseline $V(s_t)$ serving to reduce potentially high variance. The above specifies the actor-loss in the actor-critic framework. The critic is updated to minimize the advantage, i.e., $(G_t - V(s_t))^2$ is minimized. The overall algorithm with DASH-Sim is described in Algorithm 1.

---

**Algorithm 1** Deep Resource Management

---

**Input:** `jobs`, `resource_matrices`, and DASH-Sim environment
**for** each episode **do**
    initialize environment with next `job` and `resource_matrix` file
    **repeat**
        **for** tasks in ready list **do**
            Construct $state$
            Choose action $action$ w.r.t. task
            Assign $action$ to `PE_ID` for this task
            Save $state$, $action$
        **end for**
        Penalize -1 for $reward$
    **until** all tasks are the in the `completed_list` or `max_simulation_length`
    Compute losses using Eq. 1 with saved $state$s and $action$s
    Update agent by backpropagating with the losses
**end for**

---

## 4. Experiments

To the best of our knowledge, this paper is the first to apply reinforcement learning to heterogeneous resource management where long-term scheduling decisions need to be made. In this section, we show the experimental results of the comparison of the DRM scheduler with other heuristic schedulers, Earliest Finish Time (EFT), Earliest Time First (ETF), and Minimum Execution Time (MET) (Buttazzo, 2011). Both EFT and ETF pick the resource which is expected to give the earliest finish time for the task. ETF additionally schedules the task with the lowest earliest finish time first. MET assigns the processing element ID with minimum execution time for the current task.

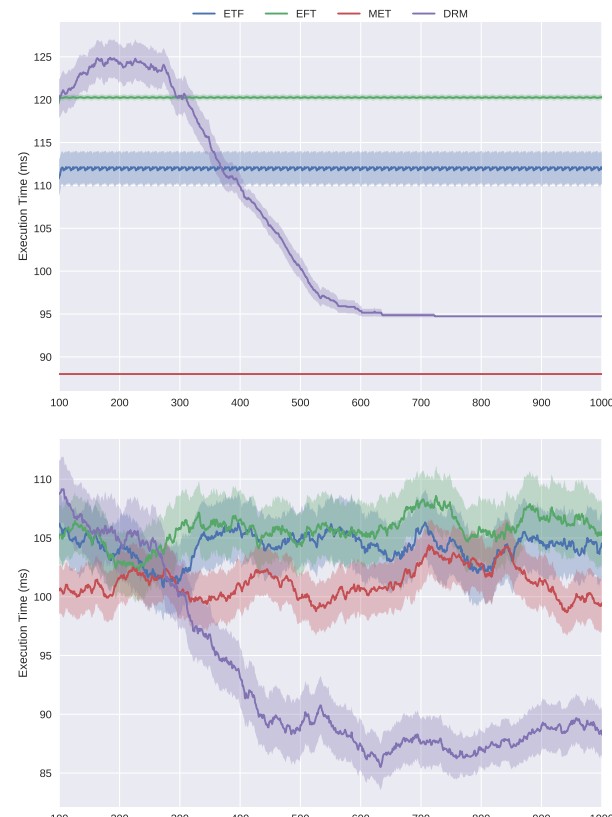

*Figure 4.* Execution Time versus Episode during training for the simpler case where the resource matrix specifying performance and communication characteristics is fixed (**Top**) and is randomized before each run (**Bottom**)

As shown in Figure 4, the deep RL scheduler has the best performance overall. We tried different experiment settings: one with fixed data shown in top and another with randomized data in bottom. According to the fixed input, the DRM agent is trained to have about 94ms performance and saturated starting at about 720 episodes. Because the static data does not have much variation, the agent does not have much variance in performance but eventually overfits to a certain extent. Interestingly, MET has better performance than DRM agent, because MET picks a resource which has the minimum execution time for the tasks in ready list. We presume the MET corresponds to a locally optimal action at every timestep whereas DRM could not exceed this optimal value.

When we experimented with the randomized data, our DRM scheduler had the best performance. Despite fluctuating results of all schedulers, the DRM agent is the only one that had improved performance, of course, over time as learning progressed. The DRM agent applies an RL algorithm to ex-

plore various policies given different jobs and PEs, allowing for better generalization and better adaptivity. To provide convincing results, we performed 30 trials with different random seeds. We expect to apply ensembles in the training experiences to provide much more reliable model.

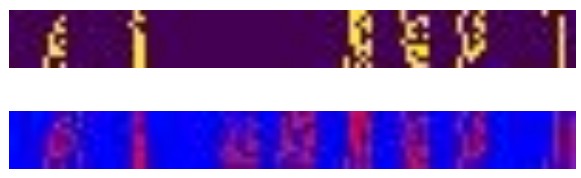

*Figure 5.* A visualization of the state representation which has information about task lists and PEs. Initialized state representation (**Top**) and the result of GradCam (Selvaraju et al., 2017) performed with the DRM feature layers (**Bottom**).

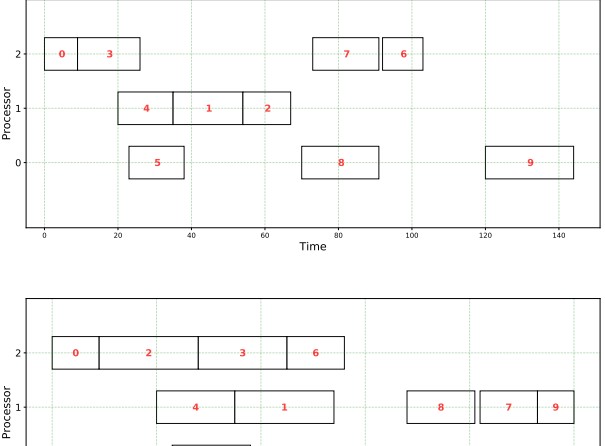

*Figure 6.* GANTT chart representing when tasks ran on the different processing elements for the first episode (Top) and the last episode (Bottom) when training DRM

We provide a visualization of the saliency to reason about the action decisions, shown in Figure 5. The top figure shows the initial state representation formed by the task lists and resource matrix information. After passing the state to DRM agent, we perform GradCam (Selvaraju et al., 2017) and retrieve the saliency, mapped onto the input, shown in the bottom of Figure 5. Notice that the agent oversees tasks which are not shown in initial representation. Described with different color intensities, we presume that the DRM agent actually understand the tasks belonging to different status lists and that this more complex decision making input allows for better policies.

Finally, the GANTT chart showcases how the policy improves over training for the fixed resource matrix case. Initially, DRM gets quite a high execution time of 140 ms while it produces a better policy of about 100 ms at the end of training. Note that this chart corresponds to the task dependency graph shown earlier, Figure 2, with the only difference being that the tasks are 0-indexed.

Some interesting changes include the choice of processing element 1 for Task 9 over processing element 0. It is cleat that this task is faster on PE1 compared to PE0. On the other hand, the choice of PE2 for task 2 vs PE1 is also interesting as task 2 takes longer. However, this might be better due to the task-dependency graph.

## 5. Related Work

Resource management has been actively researched in many communities. Several works have been applying deep RL to optimally allocate resources and distribute tasks. DeepRM uses standard deep Q-learning algorithm to formalize resource management as a Tetris game, however, it only work with homogeneous settings (Mao et al., 2016). A variant of DeepRM leverages convolution neural networks as a backbone network to improve performance in scheduling (Chen et al., 2017). Subsequent work in DeepRM, Pensieve applies a resource managing algorithm to video streaming to optimally control the bitrate and successfully reduced buffering (Mao et al., 2017). Hopfield networks have been applied to schedule heterogeneous SoC (Chillet et al., 2011). More recent work combines heuristic and learning algorithms, starting from an existing plan and iteratively improving it and successfully applying it in heterogeneous job scheduling task (Chen & Tian, 2018). However, their work follows the general MDP setting where, again, the agent chooses action at every timestep. From the perspective of hardware, recent work has proposed new accelerator architectures which have potential advances (Chen et al., 2018).

## 6. Conclusion

Neural schedulers using deep reinforcement learning have been researched in many areas and greatly improved the performance compared to heuristic algorithms. In this paper, we propose the promising approach of resource allocation applied in heterogeneous SoC chips. We use 'no-operation' action and refer to all task lists, regardless of task status, to address partially-observability and the SMDP problem. To the best of our knowledge, this paper has been the first to deal with scheduling different tasks on different hardware chips to discover the optimal combination of functionalities. Furthermore, we expect the general value functions and predictive knowledge approach and the option framework to improve performance and leave them as future work.

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
