# OpenReview forum: "Neural Heterogeneous Scheduler"
_ICML.cc/2019/Workshop/RL4RealLife — Submitted to RL4RealLife 2019_

### Official Review · AnonReviewer2 · 2019-05-16
**Interesting problem, seems to have a lot of work, but the writing needs to improve a lot**

**Rating:** 2
**Confidence:** 4

**Review:**

It is interesting to apply RL to heterogeneous scheduling problem. Getting RL to work on job scheduling has many challenges. This paper takes a non-trivial step in this direction and has some interesting results. However, the clarity of the core  technical concepts of this paper needs to improve a lot.

Detailed comments:

The paper deals with jobs with dependency structures. How does this work explicitly encode this structure? Section 3.2 and 3.1.2 only talks about implementations for the representation of jobs to a program. But I find it very hard to understand how an RL agent can process the dependencies (such as the DAG in figure 2).

The background section talks about SMDP. I’m confused why it is necessary to introduce this concept. Taking intermedia actions without a reward can still be an MDP (state transition does not depend on the past). Also, the “no-operation” action is very hard to understand.

In 3.1.2, the tasks are constantly generated. This makes it an infinite-horizon MDP. Does this fit into the standard episodic actor critic model in algorithm 1?

In section 4, the author claims that “this paper is the first to apply RL to heterogenous resource management”. However, Mirhoseini et al. applied RL to device placement problem that involves jobs with dependencies and heterogenous compute resource (different CPUs and GPUs). These two papers are worth reviewing: Device Placement Optimization with Reinforcement Learning (NIPS, 2017) and A Hierarchical Model for Device Placement (ICLR, 2018). Also, “Learning scheduling algorithms for data processing clusters” by Mao et al. applied RL to Spark cluster scheduling with general job dependency structure.

In section 5, “DeepRM uses standard deep Q-learning algorithm to …” is incorrect. DeepRM uses a modified version of REINFORCE (similar to algorithm 1 described in this paper).

---

### Official Review · AnonReviewer1 · 2019-05-24
**Bordeline. Good problem. Missing technical details and references. Limited empirical evaluation.**

**Rating:** 3
**Confidence:** 4

**Review:**

This paper proposes a reinforcement learning scheduler, which is implemented using a neural network. The scheduler decides adaptively which jobs to execute. The job can be executed only if all jobs that this job depends on have been executed. The goal is to minimize the total execution time of a batch of jobs. This problem is challenging because of potentially complicated dependencies between the jobs.

I like the problem in this paper. It is general and definitely important. This paper suffers from several issues though. My detailed comments are below:

1) Many technical details are missing. For instance, the jobs are scheduled conditioned on the past, which is represented by a feature vector. The construction of the feature vector is described in Section 3.2. This description is vague and it is unclear what the feature vector is. The optimization step in (1) is also unclear. How are A and \pi represented? Can you provide more details on how you optimize them?

2) Missing references. When you talk topics, such as actor-critic in Section 3.2, cite relevant papers, such as

  https://papers.nips.cc/paper/1786-actor-critic-algorithms.pdf

3) Limited empirical evaluation. The class of problems in Figure 2 is very limiting and does not do justice to the generality of your approach.

---

### Decision · Program_Chairs · 2019-05-28

Reject